# Decoding Pulmonary Embolism: Pathophysiology, Diagnosis, and Treatment

**DOI:** 10.3390/biomedicines12091936

**Published:** 2024-08-23

**Authors:** Miriam Peracaula, Laura Sebastian, Iria Francisco, Marc Bonnin Vilaplana, Diego A. Rodríguez-Chiaradía, Olga Tura-Ceide

**Affiliations:** 1Translational Research Group on Cardiovascular Respiratory Diseases (CAREs), Dr. Josep Trueta University Hospital de Girona, Santa Caterina Hospital de Salt and the Girona Biomedical Research Institute (IDIBGI-CERCA), 17190 Girona, Spain; mperacaula@idibgi.org; 2Department of Medical Sciences, Faculty of Medicine, University of Girona, 17003 Girona, Spain; lsebastian.girona.ics@gencat.cat (L.S.); mbonnin.girona.ics@gencat.cat (M.B.V.); 3Department of Pulmonary Medicine, Dr. Josep Trueta University Hospital de Girona, Santa Caterina Hospital de Salt, and the Girona Biomedical Research Institute (IDIBGI-CERCA), 17190 Girona, Spain; 4Department of Internal Medicine, Dr. Josep Trueta University Hospital de Girona, 17190 Girona, Spain; ifrancisco.girona.ics@gencat.cat; 5Pulmonology Department-Muscle Wasting and Cachexia in Chronic Respiratory Diseases and Lung Cancer Research Group, IMIM-Hospital del Mar, Parc de Salut Mar, Department of Medicine and Life Sciences (MELIS), Universitat Pompeu Fabra (UPF), Barcelona Biomedical Research Park (PRBB), 08003 Barcelona, Spain; darodriguez@psmar.cat; 6Biomedical Research Networking Centre on Respiratory Diseases (CIBERES), 28029 Madrid, Spain; 7Department of Biological Sciences, Faculty of Science, University of Girona, 17003 Girona, Spain

**Keywords:** pulmonary embolism, endothelial dysfunction, vascular diseases, diagnostic biomarkers, thrombolytic therapy

## Abstract

Pulmonary Embolism (PE) is a life-threatening condition initiated by the presence of blood clots in the pulmonary arteries, leading to severe morbidity and mortality. Underlying mechanisms involve endothelial dysfunction, including impaired blood flow regulation, a pro-thrombotic state, inflammation, heightened oxidative stress, and altered vascular remodeling. These mechanisms contribute to vascular diseases stemming from PE, such as recurrent thromboembolism, chronic thromboembolic pulmonary hypertension, post-thrombotic syndrome, right heart failure, and cardiogenic shock. Detailing key risk factors and utilizing hemodynamic stability-based categorization, the review aims for precise risk stratification by applying established diagnostic tools and scoring systems. This article explores both conventional and emerging biomarkers as potential diagnostic tools. Additionally, by synthesizing existing knowledge, it provides a comprehensive outlook of the current enhanced PE management and preventive strategies. The conclusion underscores the need for future research to improve diagnostic accuracy and therapeutic effectiveness in PE.

## 1. Pulmonary Embolism (PE)

Pulmonary embolism (PE) is a critical and potentially life-threatening condition that poses a significant challenge in modern medicine. It is characterized by the obstruction of blood flow in the pulmonary arteries or their branches, primarily due to thrombotic material, resulting in substantial morbidity and mortality [1,2,3,4]. Typically, obstruction is caused by the translocation of a thrombus that originates elsewhere in the body, frequently, from deep vein thrombosis (DVT) in the lower extremities [5,6,7]. Between 39 to 115 per 100,000 individuals suffer from PE each year [8,9]. Globally, PE is the third leading cause of cardiovascular-related deaths, after coronary artery disease and stroke [10]. However, estimating the precise case/fatality rates for PE can be challenging, as many patients experiencing sudden cardiac death are believed to have had an associated thromboembolic event such as PE [11,12]. While progress in diagnostic methods, risk evaluation, and treatment alternatives have improved our capacity to manage PE, considerable challenges remain in achieving prompt and precise diagnoses, refining therapeutic approaches, and minimizing long-term complications [1,2,3].

The risk factors associated with PE closely correspond with those of DVT and encompass a broad spectrum of genetic, clinical, and acquired factors. Thrombophilia constitutes an exemplar of genetic risk factors, while the most prevalent clinical factors [13] include a history of previous venous thromboembolism and obesity [14,15,16,17,18]. Other risk factors include extended periods of inactivity, such as immobilization of the lower limbs, recent hospitalization for medical or surgical reasons, extended travel lasting more than 4 h, the presence of indwelling venous catheters or central venous lines, malignancies, pregnancy, chemotherapy, hormone replacement therapy, and the use of oral contraceptive pills, among others [5,6,7,19,20].

PE is categorized into two main groups dependent on the presence or absence of hemodynamic stability. The first group, 1-Hemodynamically unstable PE (previously referred to as massive or high-risk PE) is characterized by a drop in systolic blood pressure (SBP) below 90 mmHg, a decrease in SBP by 40 mmHg or more from baseline, or the need for vasopressors or inotropes to manage blood pressure. It is important to note that the term “massive” PE does not describe the physical size of the embolism but rather its significant impact on hemodynamics. Patients with hemodynamically unstable PE have a higher risk of mortality due to obstructive shock, often linked to severe right ventricular failure [13,21].

The second group is 2-Hemodynamically stable PE (previously categorized as non-massive or low-risk), which encompasses a diverse range of events, including cases of low-risk symptomatic or asymptomatic PE and cases where PEs result in mild hypotension that can be addressed and stabilized through fluid therapy. Patients presenting with right ventricle (RV) dysfunction, while still maintaining hemodynamic stability, are categorized as being at intermediate risk of PE [13,21,22]. Based upon the European Society of Cardiology (ESC) guidelines, these patients have been further categorized as intermediate–high and intermediate–low. Intermediate–high-risk patients present with both abnormal right ventricular function and elevated serum troponin T, whereas the intermediate–low-risk group shows either abnormal right ventricular function or elevated serum troponin T [4]. 

This review aims to comprehensively examine the current state of knowledge of PE, with a focus on its pathophysiology, clinical presentation, diagnostic approaches, and therapeutic interventions. By synthesizing the latest research findings, clinical guidelines, and emerging trends, this review seeks to provide a holistic perspective on PE that can inform clinicians, researchers, and healthcare stakeholders in their efforts to enhance patient outcomes and reduce the burden of this critical cardiovascular disorder. We also explore future directions and potential areas for further investigation in the field of PE management and prevention.

## 2. Diagnosis and Severity

Identification of DVT and PE involves a series of diagnostic techniques, which should be applied in a sequential manner. Early diagnosis is crucial, especially in cases of massive PE typically characterized by clinical signs of shock or hemodynamic instability [23]. Following a standardized diagnostic pathway is highly recommended because it has demonstrated a significant reduction in the risk of complications. To achieve this, it is essential to stratify patients based on their probability of developing a PE.

Clinical probability assessment is used to identify patients exhibiting possible symptoms of PE, helping to discern those with high or intermediate clinical probabilities of suffering a PE. Clinical probability assessment includes evaluating clinical factors and predisposing factors (Table 1) [24,25,26]. These probabilities can be determined empirically or by using prediction rules or scores. This aids in determining the need for anticoagulant treatment while awaiting the results of diagnostic tests.

### 2.1. Scoring Systems Diagnosis

Scoring systems are probability scores designed to help the physician to confirm or exclude PE. Two widely used scores for suspected PE are the *Wells score* and the *Geneva score* [27,28,29,30]. Both the *Wells Score* and the *Geneva Score* assign points to each clinical and patient factor (Table 2) to calculate a total score, which categorizes patients into low, intermediate, or high probability for PE. This classification aids in the selection of appropriate diagnostic tests, such as D-dimer assays and imaging studies, ensuring the efficient and accurate diagnosis of PE [31,32]. 

The choice of a particular prediction rule and classification method is influenced by factors such as the local prevalence of PE, the patient population, and the type of D-dimer assay employed. For instance, the revised *Geneva score* is recommended for populations with a prevalence of PE exceeding 20%, while the *Wells score* is the sole validated score for hospitalized patients [33]. 

### 2.2. Risk Stratification of PE

The utilization of the *Wells Score* or the *Geneva Score* facilitates the classification of patients into specific risk categories. The diagnosis of PE involves evaluating D-dimer tests, Computed tomography pulmonary angiographies (CTPAs), and ventilation/perfusion (V/Q) scans, as well as supplementary imaging techniques (Figure 1). The selection of a specific diagnostic test varies based on the patient’s categorization:Low Probability of PE: to safely exclude PE in this subset of patients, the physician can use the Pulmonary Embolism Rule Out Criteria (PERC) or D-dimer testing. Patients with a low probability of PE but a non-negative PERC rule should be tested with D-dimer to safely exclude PE.Intermediate Probability of PE: The initial diagnostic step involves the measurement of D-dimer levels. Negative D-dimer results, indicating levels below 500 ng/mL, allow for the confident exclusion of PE. However, a positive D-dimer test, defined as D-dimer levels above 500 ng/mL, indicates that CTPA should be applied next. If the CTPA results are inconclusive or contraindicative, a V/Q scan should be considered (Figure 1) [13].High Probability of PE: CTPA should be performed promptly. In situations where the CTPA yields inconclusive results, a V/Q scan is recommended. The V/Q scan findings may either be normal, effectively ruling out PE, or they may indicate a “high probability of PE”, which is considered diagnostic for PE. When the V/Q scan results fall into the intermediate probability range, further assessment with lower extremity compression ultrasonography (CUS) is warranted (Figure 1) [13].

## 3. Treatment

The primary approach to treating PE involves preventing further clot formation and managing associated symptoms. The most common treatments in PE are anticoagulation therapy and thrombolytic therapy. 

### 3.1. Conventional Anticoagulation Therapy

Anticoagulants such as heparin, direct oral anticoagulants (DOACs), and vitamin K antagonists (VKAs) are commonly prescribed to inhibit the formation of new blood clots and prevent the extension of existing clots in different moments. Heparins and DOACs are indicated in the acute phase, and DOACs and VKAs are suggested for the subacute phase of the disease [4]. 

However, in the acute phase of pulmonary embolism (PE), intravenous non-fractionated heparin or subcutaneous low-molecular weight heparin (LMWH) are typically prescribed due to their rapid onset of action and proven efficacy. After the acute phase, long-term anticoagulation therapy is required to prevent the recurrence of PE. For this purpose, VKAs and DOACs are often recommended at discharge [34].

Heparin, a glycosaminoglycan, acts as an indirect anticoagulant by enhancing antithrombin III activity. When activated by heparin, antithrombin III effectively inhibits key coagulation factors, including thrombin (factor IIa) and factor Xa (Figure 2). Thrombin is pivotal in clot formation by converting fibrinogen to fibrin. By inactivating thrombin and factor Xa, heparin reduces clot formation, indirectly supporting endothelial function and preventing vascular obstruction [35]. There are three main types of heparins: standard heparin, low-molecular weight heparin (LMWH), and Fondaparinux. LMWH’s functions are similar to standard heparin by binding to and boosting the activity of antithrombin III (AT), but it mainly targets factor Xa and has a longer-lasting effect. In addition, LMWH has a reduced ability to bind and inhibit thrombin compared to standard heparin. LMWH has a smaller size which results in less binding to plasma proteins and endothelial cells, resulting in a significant amount of unbound LMWH. This leads to a high bioavailability rate of 85–99%, enhancing its anticoagulant effect, prolonging its action, and reducing patients’ variability. Fondaparinux binds to antithrombin III with high specificity, minimizing non-specific interactions with other plasma proteins. It significantly increases AT’s activity, especially in inhibiting factor Xa. Fondaparinux does not directly impact thrombin and exhibits high bioavailability when administered subcutaneously [4].Warfarin is an oral anticoagulant that hinders the synthesis of vitamin K-dependent clotting factors (II, VII, IX, and X) by inhibiting the enzyme vitamin K epoxide reductase, which recycles vitamin K (Figure 2). This disruption in clotting factor production reduces the blood’s clotting ability, indirectly supporting endothelial function by preventing new clot formation and the extension of existing clots [36].

### 3.2. Direct-Acting Oral Anticoagulants

Direct-acting oral anticoagulants (DOACs) fall into three classes: anti-Xa agents (Rivaroxaban, Apixaban, and Edoxaban) antithrombin agents (Dabigatran) [37], and anti-XI agents (ASOs). These synthetic, low-molecular weight compounds act as direct, selective, and reversible inhibitors of specific coagulation cascade steps [38,39,40,41,42,43,44]. Their anticoagulant effects are predictably consistent, eliminating the need for laboratory monitoring or dose adjustments, unlike heparin or vitamin K antagonists. The relatively short half-lives of these agents simplify clinical management for situations, such as invasive procedures or bleeding events requiring anticoagulation reversal or adjustment, [45]. 

Anti-Xa agents are highly specific direct inhibitors of factor Xa, a central enzyme in the blood coagulation cascade. It achieves its specificity by directly binding to factor Xa’s active site, thereby preventing the enzyme from carrying out its crucial function (Figure 2). By directly inhibiting factor Xa, it effectively interrupts the production of thrombin, which, in turn, prevents the formation of fibrin, a key component for the stability of blood clots [39,41,43,44].Antithrombin agents interact with antithrombin, a crucial endogenous anticoagulant protein (Figure 2). Their primary molecular role is to potentiate antithrombin’s inhibitory capacity against key coagulation factors, with particular emphasis on thrombin (factor IIa) and factor Xa. This interaction induces significant conformational changes in antithrombin, effectively boosting its ability to neutralize thrombin and factor Xa. As a result, the coagulation cascade is disrupted, and the formation of blood clots is impeded [38].

### 3.3. Anti-XI Agents

Anti-XI agents are specific inhibitors that target factor XIa within the intrinsic pathway of the coagulation cascade. By inhibiting factor XIa, they disrupt the conversion of factor IX to factor IXa, a crucial step in the activation of Factor X and subsequent thrombin and fibrin clot formation [46]. Therefore, anti-XI agents prevent blood clots similarly to DOACs but they work through different mechanisms. Anti-XI agents present a lower risk of bleeding compared to DOACs. Research highlights the significant potential of FXI inhibitors in managing low-risk and intermediate–low-risk PE and DVT. Clinical trials suggest that FXI inhibitors may offer superior efficacy and safety over conventional anticoagulants, including DOACs. This is particularly relevant for cancer-associated PE, where current treatments such as LMWH and DOACs have high drop-out rates and bleeding risks [47].

Despite their potential advantages, these anticoagulants are currently in the investigational stages and have not yet received regulatory approval for clinical use in many countries, limiting their use compared to other established treatments [47]. 

### 3.4. Thrombolytic Therapy

In severe cases of PE, systemic fibrinolytic therapy is employed to rapidly dissolve existing blood clots, relieving pressure on the pulmonary vasculature and potentially promoting endothelial function over time. Thrombolytic agents can directly or indirectly activate plasminogen, which is then transformed into plasmin. Plasmin is a potent proteolytic enzyme that targets fibrin, a critical structural component of blood clots. This process, known as fibrinolysis, results in the breakdown of clots into soluble fibrin fragments, effectively clearing clots and preventing vascular occlusion [48].

There are two main categories of thrombolytic agents. The first group includes tissue plasminogen activator (tPA) and Urokinase, which are direct plasminogen activators. tPA is a highly specific recombinant protein that functions as a serine protease, while Urokinase is a less specific endogenous human enzyme [49]. The second group comprises non-plasminogen activators, with Streptokinase being a prominent member. Streptokinase, a bacterial protein, indirectly activates plasminogen by forming a complex with it [50]. These thrombolytic agents effectively address severe PE by dissolving blood clots and mitigating life-threatening complications.

### 3.5. Thrombectomy Procedure

Thrombectomy is a surgical procedure involving the extraction of blood clots from blood vessels and is employed in instances necessitating clot removal. Within the domain of DVT or PE, thrombectomy emerges as a consideration in cases of pronounced, severe, or critical cases. Its application is particularly pertinent in scenarios characterized by massive PE, where a substantial obstruction in the pulmonary arteries gives rise to a life-threatening condition. Furthermore, thrombectomy may be contemplated when anticoagulant therapy in isolation proves insufficient to expedite clot dispersion or in situations marked by hemodynamic instability. The decision to proceed with thrombectomy is contingent upon an individualized clinical evaluation, carefully balancing the gravity of the patient’s condition with the attendant procedural risks [51]. As an invasive procedure, thrombectomy is selectively utilized when its benefits outweigh potential complications. Additionally, pulmonary artery catheterization or direct thrombolytic drug infusion therapy are catheter procedures, necessitating the insertion of a catheter into the pulmonary artery to perform specific treatment, such as the administration of thrombolytic drugs or mechanical removal of blood clots. This will help to restore blood flow to mitigate life-threatening complications [4].

These surgical interventions play a pivotal role in the management of severe thrombotic events and are indispensable in restoring blood flow for mitigating life-threatening complications [51,52]. 

#### 3.5.1. Mechanical Thrombectomy (MT)

MT involves the use of specialized devices designed to physically remove blood clots from vessels. These devices can include wire-guided aspiration catheters, which suction out clots, or mechanical retrieval systems, such as stent retrievers, which ensnare and extract clots. Mechanical thrombectomy is particularly useful in cases where clots are large, resistant to dissolution by medications, or when rapid restoration of blood flow is crucial. The technique is often employed in the treatment of massive PE, where large clots in the pulmonary arteries may cause severe hemodynamic instability or even right ventricle dysfunction. Mechanical thrombectomy can quickly reduce the clot burden and improve hemodynamic parameters with a low bleeding risk when compared to systemic thrombolysis [47]. 

#### 3.5.2. Catheter-Directed Thrombolysis (CDT)

CDT is a less invasive alternative technique that involves the targeted delivery of thrombolytic agents directly to the site of the clot via a catheter. This procedure allows for lower doses of thrombolytic drugs compared to systemic administration, reducing the risk of bleeding complications while enhancing the effectiveness of clot dissolution. By delivering medication directly into the clot, CDT accelerates clot breakdown, minimizes vessel damage, and promotes faster restoration of blood flow. CDT is particularly advantageous in treating DVT and submassive PE, where clot burden is significant but the risk of full systemic thrombolysis is considered too high [47]. 

### 3.6. Supportive Care 

Supportive care in managing PE involves the strategic use of oxygen therapy. This addresses critical aspects of patient well-being by correcting hypoxemia, a common PE complication. Oxygen therapy increases blood oxygen content, improving oxygen delivery to vital organs and tissues. It also plays a crucial role in alleviating right heart strain caused by PE, reducing the heart’s workload through enhanced oxygen saturation. Furthermore, oxygen therapy effectively manages PE symptoms, such as shortness of breath and chest pain exacerbated by low oxygen levels, ultimately enhancing patient comfort and their overall condition [51,53,54]. Individualized oxygen therapy regimens involve tailoring the supplemental oxygen administration to meet the specific needs of each patient in order to optimize oxygen levels while closely monitoring for potential side effects [55]. 

### 3.7. Multidisciplinary Pulmonary Embolism Response Teams

The definition of multidisciplinary rapid-response teams to manage cases of PE includes high-risk and specific intermediate-risk cases and emerged in the United States [4]. This approach has gained value within the medical community and has been increasingly adopted in hospitals across Europe and globally. The establishment of Pulmonary Embolism Response Teams (PERTs) is necessary due to their ability to address the requirements of modern healthcare systems [47]. A PERT comprises specialists from various fields, such as cardiology, pulmonology, hematology, vascular medicine, anesthesiology/intensive care, cardiothoracic surgery, and (interventional) radiology. These specialists convene in real-time to enhance clinical decision-making. This allows the development of a treatment plan which can be implemented immediately. The specific composition and operational procedures of a PERT are not fixed, depending on the resources and expertise available within each hospital for the management of acute PE [4]. 

## 4. Potential Biomarkers in PE

Conventional PE biomarkers result from various biological processes, including coagulation and fibrinolysis cascades, immune responses, and organ function (Table 3) [56,57,58]. In recent years, new methods have been applied to PE diagnosis, with microRNA being of particular interest as a potential biomarker. Most biomarkers can be classified into different biological categories: vascular biomarkers, cardiac biomarkers, inflammatory biomarkers, and RNAs.

### 4.1. Vascular Biomarkers of PE

Within the coagulation and fibrinolysis cascades, various blood count parameters have been examined as potential indicators of PE risk (Table 3). Elevated leukocyte and platelet counts have been associated with an increased risk of pulmonary occlusion, while the levels of D-dimer and fibrinogen have demonstrated a clear relationship with PE [59,60,61]. Furthermore, soluble P-selectin (sP-selectin), which acts as a cell adhesion molecule and an indicator of platelet and endothelial activation, has also shown promise as a good predictor of PE [62].

In addition, coagulation factors such as II, IV, VII, VIII, V Leiden, X, XIII, prothrombin fragments 1+2, antithrombin III, tissue factor, protein C, and protein S have also been linked to PE [63,64,65]. Xiong et al., demonstrated that coagulation factor IV may serve as an indicator of symptomatic PE [66]. Zhou et al. also added the thrombin–antithrombin III complex (TAT), the plasmin-α2–plasmininhibitor complex (PIC), thrombomodulin (TM), and the tissue plasminogen activator–inhibitor complex (t-PAIC), which play crucial roles in the venous thrombosis process and can be elevated before thrombus formation, as good PE biomarkers [67].

Alternatively, common blood tests such as activated partial thromboplastin time (aPTT), prothrombin time (PT), and thrombin time (TT) may exhibit limited sensitivity and specificity for PE diagnostic [68]. However, their value lies in providing a broader view of the patient’s coagulation status and guiding further diagnostic steps in suspected cases of PE. Additionally, viscoelastic hemostatic assays (VHAs) provide real-time information about fibrin formation, platelet activation, and clot retraction, making them valuable for the diagnosis and management of PE [69].

### 4.2. Cardiac Biomarkers of PE

In the field of cardiac biomarkers, troponins such as troponin I (TnI) and troponin T (TnT) serve as highly sensitive diagnostic biomarkers; elevated troponin levels predict adverse consequences in conditions such as acute myocardial infarction and critical illness. In the context of PE, normal TnT levels indicate a favorable prognosis [70,71,72]. However, it has been demonstrated that using TnT alone for predicting PE complications and assessing the risk of hemodynamically stable PE patients is insufficient. 

Recent clinical studies have shown that b-type natriuretic peptide (BNP) can increase during PE and is valuable for risk stratification and prognosis assessment. The increase in PAP and RV wall tension in PE leads to elevated BNP concentrations in the blood. Both BNP and its inactive form NT-proBNP are associated with RV dysfunction and are predictive of short-term mortality in PE patients [73]. Simultaneous detection of serum TnT and BNP has been shown to improve risk stratification and positive prognostic value in PE patients [71].

Furthermore, heart-type fatty acid binding protein (h-FABP), which is specific to myocardial injury, is released into the blood when cardiomyocytes are damaged [74,75]. In cases where PE is confirmed, BNP/NT-proBNP and h-FABP can be used together for improved risk stratification [76,77]. Additionally, some patients with PE may also exhibit increased serum creatine kinase (CK) and myoglobin [74,78].

### 4.3. Inflammatory Biomarkers of PE

Recent studies related to inflammatory biomarkers have identified new inflammatory markers closely associated with PE (Table 3), such as the neutrophil–lymphocyte ratio (NLR), platelet–lymphocyte ratio (PLR), and lymphocyte–monocyte ratio (LMR) [79]. In particular, the NLR emerges as an independent predictor of in-hospital mortality in acute PE patients and has been useful for clinical risk stratification. Additionally, elevated red cell distribution width (RDW) is also considered a biochemical marker of pre-inflammatory conditions and serves as a prognostic indicator in PE patients [80,81]. Moreover, several studies have linked anticardiolipin antibodies such as IgG and IgA to PE [74,78].

In this direction, Araz et al. demonstrated that high levels of serum C-reactive protein (CRP), which is synthesized by the liver and is released into circulation in response to inflammation, correlate with the severity of PE [82].

While it is important to note that monitoring of inflammatory markers is not currently recommended by clinical practice guidelines, recent studies suggest potential utility. Elevated plasma levels of various cytokines, including interleukins (ILs) and TNF-α, in PE patients indicate the significance of inflammation in PE. In addition, elevated TNF-α is particularly sensitive for predicting lung injury from PE, while IL-1β, IL-4, IL-6, and hs-CRP can be used to monitor PE occurrence and guide and assess the efficacy of treatment [83,84]. Liu et al. reported that IL-10 aids in reducing lung injury and facilitating tissue repair [85]. Additionally, Ozmen et al. showed that plasma copeptin levels reflect PE severity and can be useful for prognostic assessment [86]. While these findings show promise, it is crucial to acknowledge that further research and validation are needed to establish their clinical validity and integration into standard practice guidelines.

### 4.4. RNA Biomarkers of PE

For considering RNA-based biomarkers, it is essential to recognize the roles of various non-coding RNAs in PE such as MicroRNAs, lncRNA, and circular RNA.

MicroRNAs (miRNAs) are non-coding small RNA molecules with specific tissue expression patterns. Their expression changes significantly in some diseases, especially thrombosis and tumor diseases [87]. Certain miRNAs such as miR-210, miR-221, miR-222, miR-126-3p, miR-92a, and miR-132, which are known as angiomiRs (Table 3), have an essential function in regulating angiogenesis. Hembron et al. have reported more novel biomarkers, such as miR-223, miR-145, miR-582, miR-195, miR-150, miR-21, and miR-424, which are related to the changes in vascular diseases [88]. In addition, other studies have identified miR-134 [89], miR-28-3p [90], and miR-1233 [91] as potential biomarkers for PE diagnosis.Long non-coding RNAs (lncRNAs) are non-coding RNA molecules longer than 200 nucleotides. Some studies have shown that lncRNAs have significant level changes in chronic thromboembolic pulmonary hypertension (CTEPH) (Table 3), such as NR_036693, NR_027783, NR_033766, and NR_001284 [92,93]. Additionally, LncRNA-Ang362 has also been closely linked to PH [92,94]. Further studies are needed to determine the specific connection between lncRNAs and PE.Circular RNAs (circRNAs) are closed-loop RNA molecules formed by back-splicing events. Various studies have reported different circRNAs such as hsa_circ_0002062, hsa_circ_0022342, hsa_circ_0016070, and hsa_circ_0046159, which are implicated in CTEPH and PH (Table 3) [95,96,97].

## 5. Endothelial Dysfunction in PE

Endothelial dysfunction (ED) plays a critical role in various vascular diseases, including PE. The endothelium is a monolayer of endothelial cells lining blood vessels and the lymphatic system [98,99]. Its functions are diverse, regulating blood fluidity, platelet aggregation, vascular tone, immunological responses, inflammation, and angiogenesis, and can also act as an endocrine organ [98,99]. Maintaining endothelial homeostasis is essential for overall vascular health, as an alteration in the endothelium can contribute to the development of vascular diseases [98].

The endothelium can undergo structural or functional changes due to various clinical risk factors, including genetic predisposition, recent surgery, prior thromboembolic events, and hemoptysis, among others. Additionally, cellular risk factors such as oxidative stress, metabolic alterations, and inflammation can trigger decreased endothelial vasodilator capacity, pro-thrombotic responses, and abnormal modulation of vascular growth, potentially leading to pulmonary embolism (PE) (Figure 3) [100,101]. 

## 6. Impact of PE in ED

PE can also potentially disrupt the normal function of ECs, leading to endothelial injury. This can be attributed to multiple factors, including the presence of a blood clot in the pulmonary arteries and the release of inflammatory mediators. The impaired vasodilator capacity of the endothelium in PE patients can contribute to increased pulmonary vascular resistance, potentially leading to RV dysfunction. Moreover, the pro-inflammatory and pro-thrombotic responses of the endothelium can exacerbate the formation of blood clots within the pulmonary vasculature, further obstructing blood flow and affecting overall vascular health. Additionally, abnormal modulation of vascular growth in PE may impact the remodeling of pulmonary blood vessels, which is an important aspect of adapting to the changes induced by PE [102]. 

### 6.1. Impaired Blood Flow Regulation 

The endothelium’s precise regulation of vasoconstriction and vasodilation is critical for maintaining vascular tone. ED in PE disturbs this balance, leading to an increase in pulmonary vascular resistance (PVR). This occurs due to the diminished production and release of vasodilatory factors, such as nitric oxide (NO), and an increased presence of vasoconstrictors such as endothelin-1 (Figure 3) [103]. The consequence is constriction of pulmonary blood vessels, increasing PVR and placing a greater workload on the RV. These events lead to an increased risk of developing vascular diseases. 

### 6.2. Pro-Thrombotic State

ED disrupts the endothelium’s anticoagulant properties, characterized by reduced synthesis and release of anticoagulant molecules such as tissue factor pathway inhibitor (TFPI) and heparin sulfate (HS), coupled with an increase in procoagulant factors, including tissue factor (TF) and von Willebrand Factor (vWF). This imbalance in the coagulation cascade favors clot formation, heightening the risk of thrombus development in the pulmonary vasculature, a hallmark of PE (Figure 3) [104,105].

### 6.3. Inflammatory State

When the endothelium is damaged, it leads to the release of pro-inflammatory molecules, such as interleukin-1 (IL-1), interleukin-6 (IL-6), tumor necrosis factor alpha (TNF-α), and C-reactive protein (CRP). This pro-inflammatory state contributes to ED (Figure 3). Additionally, a dysfunctional endothelium is marked by increased expression of cell adhesion molecules (CAMs), including E-selectin, vascular cell adhesion molecule-1 (VCAM-1), and intercellular adhesion molecule 1 (ICAM-1), stimulated by activation of nuclear transcription factor NF-kB when cytokines bind to their receptors [106,107]. Soluble forms of CAMs can also circulate in the bloodstream due to endothelial cell activation and secretion [107,108]. 

These CAMs promote leukocyte adhesion and migration to the damaged areas in order to repair the affected tissue. However, abnormally increased levels of CAMs can cause excess or prolonged leukocyte adhesion or migration, leading to chronic inflammation which is associated with pathological conditions such as PE [106,107].

### 6.4. Increased Oxidative Stress 

ED often correlates with elevated oxidative stress, characterized by an imbalance between the generation of reactive oxygen species (ROS) and the endothelium’s antioxidant defenses. This imbalance results in excess ROS, disrupting redox balance within endothelial cells and activating pro-inflammatory pathways such as NF-kB. 

This sets off a chain reaction involving CAMs, also activating the overall cascade of adhesion and pro-inflammatory molecules. 

Oxidative stress directly damages endothelial cells, leading to lipid peroxidation, protein oxidation, and DNA damage, contributing to vascular injury and impaired endothelial function. Additionally, oxidative stress can hinder the production and availability of NO, converting it into harmful peroxynitrite. The reduction of NO can also be attributed to multiple other factors, including elevated inflammatory CRP which can downregulate the expression and bioactivity of endothelial nitric oxide synthase (eNOS), a key enzyme responsible for NO production. Reduced NO bioavailability impairs vasodilation, upsetting the balance of vasoconstriction and vasodilation, a significant concern in the context of PE (Figure 3) [107,109].

### 6.5. Altered Vascular Remodeling

Prolonged ED can instigate a process of vascular remodeling. This involves structural changes in the pulmonary vasculature resulting in vascular wall thickening, fibrosis, and increased muscularization of small pulmonary arteries (Figure 3). Enhanced smooth muscle cell proliferation and contraction, endothelial cell dysfunction, and extracellular matrix remodeling accompany these adaptations [110]. This vascular remodeling can cause pulmonary hypertension (PH), defined as an increase in PVR and pulmonary arterial pressure (PAP). This condition has significant implications for RV function and can lead to RV hypertrophy and heart failure. In addition, severe RV dilation can also lead to heart failure with preserved ejection fraction [111].

## 7. From PE to Cardiovascular Dysfunction

Recurrent venous thromboembolism (VTE) events are a common concern associated with PE. In some cases, the first VTE event goes undiagnosed, while in others, new VTE events occur after discontinuing anticoagulation treatment. In addition, PE can also lead to a post-thrombotic syndrome, recurrent thromboembolism, right heart failure, cardiogenic shock, and chronic thromboembolic pulmonary hypertension (CTEPH) [112,113,114].

### 7.1. Post-Thrombotic Syndrome (PTS)

PTS is a chronic condition that develops following an episode of DVT, which can lead to PE and its risk increases with the number of recurrent thrombotic events. Indeed, PTS arises as a long-term consequence [115] for one-third of people with PE/VTE [116]. 

PTS symptoms involve persistent leg discomfort, including pain, swelling, heaviness, and skin changes, such as brown pigmentation or, in severe cases, venous ulcers [112,113,114]. The syndrome develops due to the vein damage (inflammation, scarring, and disruption of normal blood flow) incurred during the initial DVT, leading to chronic venous insufficiency. Anticoagulation therapy and early mobilization are used in DVT patients as preventive measures to mitigate the risk of developing PTS. Managing PTS may include compression stockings, lifestyle adjustments, and, in certain situations, interventional procedures to address impaired veins [117]. 

### 7.2. Chronic Thromboembolic Pulmonary Disease

After a pulmonary embolism (PE), approximately 50% of patients report ongoing dyspnea and 30–50% have perfusion defects. Persistent symptoms, regardless of the presence or absence of chronic PE on imaging, are referred to as post-PE syndrome [4]. If imaging confirms chronic PE, the condition is termed chronic thromboembolic pulmonary disease (CTEPD). CTEPD is diagnosed based on hemodynamic criteria, including pulmonary vascular obstruction and a mean pulmonary artery pressure below 25 mmHg at rest. It is distinct from new or recurrent PE and should not be confused with them. Therefore, CTEPD is defined by the presence of chronic thromboembolic material in the pulmonary arteries without pulmonary hypertension at rest [118].

Genetic predispositions, such as coagulation disorders or comorbidities, as well as active neoplasms, may predispose some PE patients to develop CTEPD. It is crucial for PE patients to remain vigilant and promptly report any concerning symptoms such as chest pain, shortness of breath, or hemoptysis. CTPA or V/Q scans are commonly used as diagnostic tests to confirm the presence of CTEPD [119].

### 7.3. Chronic Thromboembolic Pulmonary Hypertension 

Chronic thromboembolic pulmonary hypertension (CTEPH) is a long-term consequence for 2–4% of PE patients and is characterized by persistently elevated PAP at least three months after initial PE diagnosis and effective anticoagulation treatment. This underscores the importance of ongoing medical care and monitoring for individuals who have experienced a PE, even after the acute event has been successfully managed [120].

The mechanisms leading to CTEPH are complex and not entirely understood, but they involve the inadequate resolution of blood clots in the pulmonary arteries, leading to the development of organized fibrotic material that obstructs the blood vessels and increases pulmonary artery pressure. CTEPH also acts as a mechanism of vascular remodeling independently of inadequate thrombus resolution [121]. 

Early recognition, diagnosis, and management of CTEPH are essential for improving patient outcomes and preventing complications. Debilitating symptoms, particularly breathlessness, are key indicators of CTEPH, which can lead to disabling dyspnea, reduced life expectancy, and even sudden cardiac death due to right ventricular failure.

### 7.4. Right Heart Failure

Right heart failure is a critical condition frequently associated with acute PE. It develops when the right side of the heart is unable to effectively pump blood into the pulmonary arteries and subsequently into the lungs. This condition is primarily due to increased pressure within the pulmonary arteries, a common outcome of PE which can obstruct the pulmonary arteries, diminishing blood flow to the lungs for oxygenation. Consequently, the RV has to generate elevated pressure to overcome this obstruction, resulting in increased strain and potential damage to the right side of the heart [122]. Common symptoms include breathlessness, peripheral edema, often observed as swelling in the legs and ankles, abdominal discomfort, fatigue, and heart palpitations. If not promptly treated, right heart failure can lead to severe complications such as cardiogenic shock [123].

Typical treatments are anticoagulation therapy to prevent further blood clot formation and interventions to remove or dissolve the existing blood clot. Anticoagulation treatment is only applicable when RV failure is secondary to PE and is not a treatment for RV failure from other medical causes. In some cases, surgical procedures or catheter-based interventions may be necessary to address the clot [124]. 

### 7.5. Cardiogenic Shock 

Cardiogenic shock is a critical condition characterized by the heart’s inability to maintain effective blood circulation, resulting in systemic oxygen deficiency. This life-threatening state can manifest in various clinical contexts, including as a complication of PE [125]. Clinical manifestations associated with cardiogenic shock in the context of PE include profound dyspnea, hypotension, arrhythmias, cold and clammy skin, and, in severe cases, cyanosis.

In PE patients, the development of cardiogenic shock primarily arises from the significant strain imposed on the RV of the heart, which is forced to work harder to overcome the obstruction caused by PE, leading to increased pressure and stress on the right side of the heart [126]. In acute PE, this can result in cardiogenic shock. Coronary revascularization, inotropes, and vasopressor therapy are the current management options for cardiogenic shock [127]. 

## 8. Conclusions and Future Directions

Medical intervention for the treatment of PE is currently a complex challenge, demanding a comprehensive understanding of its pathophysiology, diagnostic methodologies, and therapeutic interventions. Despite the notable progress in diagnostic methods, risk assessment, and treatment options, PE continues to present significant morbidity and mortality risk limitations in the current understanding and management of PE. The diverse clinical manifestations in PE mean that prompt and precise diagnosis continues to be challenging. Risk stratification tools reveal variations in sensitivity and specificity across different patient populations. Additionally, biomarker research shows promise but needs further validation and standardization.

The existing therapeutic strategies, including anticoagulation and thrombolytic therapies, present limitations emphasizing the need for personalized risk–benefit evaluations. Furthermore, our understanding of the complex interplay between ED and PE remains unclear, making the development of targeted therapies designed to preserve endothelial homeostasis difficult.

To address these limitations, innovative research is needed. Integration of artificial intelligence in diagnostic algorithms and RNA-based biomarkers may provide more precise insights into PE pathophysiology and prognosis. Additionally, more molecular studies are needed to elucidate underlying ED mechanisms in PE and to achieve novel treatments targeted to endothelial preservation. Finally, prospective studies investigating long-term outcomes are imperative to comprehend the full spectrum of associated vascular diseases and refine post-PE management strategies.

## Figures and Tables

**Figure 1 biomedicines-12-01936-f001:**
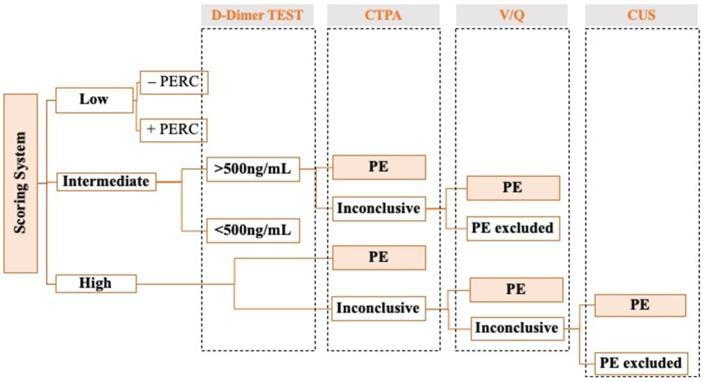
Diagnostic Algorithm for pulmonary embolism (PE) depending on the patient’s risk category score using the Geneva Clinical Prediction or Wells Criteria. Abbreviations: Pulmonary Embolism Rule Out Criteria (PERC), Pulmonary embolism (PE), Computed tomography pulmonary angiography (CTPA), ventilation/perfusion (V/Q), and extremity compression ultrasonography (CUS).

**Figure 2 biomedicines-12-01936-f002:**
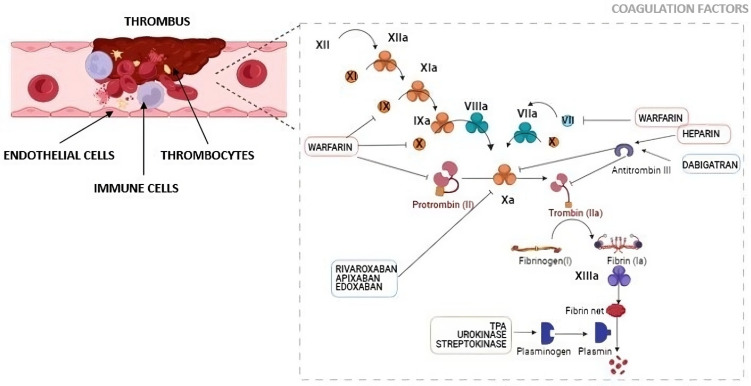
Schematic Representation of Coagulation and Clotting Factor Signaling in Thrombus Formation. The cascade involves key coagulation factors such as XII, XIIa, IXa, VIIIa, VIIa, Xa, and XIIIa, which play pivotal roles in the propagation of clotting events. The principles of the coagulation cascade pathways are detailed, highlighting the coordinated activation of these cellular components. The anticoagulant treatments for PE are detailed, encompassing conventional anticoagulants, novel oral anticoagulants, and thrombolytic agents.

**Figure 3 biomedicines-12-01936-f003:**
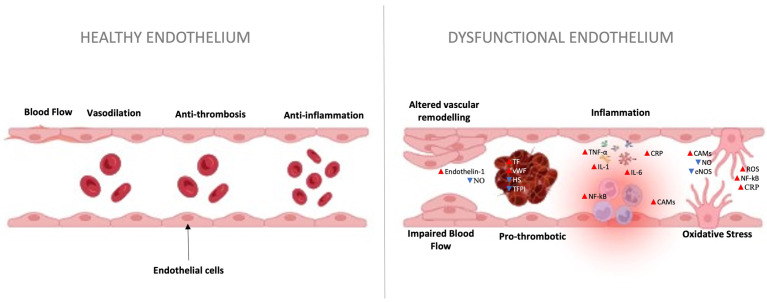
Effects of pulmonary embolism (PE) in the vascular endothelium. PE disrupts normal endothelial function, which can be attributed to multiple factors: vascular remodeling alterations, vasoconstrictor–vasodilator imbalance, anticoagulant levels and procoagulant factors, synthesis of pro-inflammatory molecules, and an increase in oxidative stress, among others. Abbreviations: Nitric oxide (NO), Tissue factor pathway inhibitor (TFPI), heparin sulfate (HS), Tissue factor (TF), von Willebrand Factor (vWF), interleukin-1 (IL-1), interleukin-6 (IL-6), tumor necrosis factor alpha (TNF-α), C-reactive protein (CRP), cell adhesion molecules (CAMs), Nuclear Factor-kappa B (NF-kB), Reactive oxygen species (ROS), and endothelial nitric oxide synthase (eNOS).

**Table 1 biomedicines-12-01936-t001:** Most common clinical symptoms and risk parameters of clinical probability Assessment.

Clinical Features	Predisposing Factors
Chest pain Dyspnea Unilateral leg swelling Syncope Hemoptysis Heart rate < 100/min Oxygen saturation < 94%	Age < 50 Familial history of PE ^1^ Familial history of cardiovascular diseases Recent surgery Prior venous thromboembolism event Estrogen use

^1^ Abbreviations: Pulmonary embolism (PE).

**Table 2 biomedicines-12-01936-t002:** Clinical decision rule points of the *Geneva* Clinical Prediction Rules (simplified version) and the modified *Wells* Criteria Rules.

*Geneva* Clinical Prediction Rules	Score	*Wells* Criteria Rules	Score
Previous PE or DVT ^1^	1	Previous PE or DVT	1.5
HR: 75–94 beats per minute	1	Heart rate >100 beats per min	1.5
HR: ≥95 beats per minute	2	Immobilization for ≥3 days or surgery in the previous 4 weeks	1.5
Surgery or fracture within the past month	1	Hemoptysis	1
Hemoptysis	1	Malignancy	1
Active cancer	1	Clinical symptoms of DVT	3
Unilateral lower-limb pain	1	Other diagnoses less likely than pulmonary embolism	3
Pain on lower-limb deep palpation and unilateral edema	1		
Age >65 years.	1		

PE—unlikely	0–2	PE—unlikely	≤4
PE—likely	≥3	PE—likely	>4

Or		Or	

Low	0–3	Low	<2
Intermediate	4–10	Intermediate	2–6
High	>10	High	>6

^1^ Abbreviations: Pulmonary embolism (PE), Deep vein thrombosis (DVT).

**Table 3 biomedicines-12-01936-t003:** List of established biomarkers for the development of pulmonary embolism (PE) and/or future vascular diseases derived from PE and their medical use.

General Types of Biomarkers	Potential PE Biomarkers	Medical Uses
Vascular biomarkers	*Leukocyte*, *platelet*, *D-dimer*, *fibrinogen*, *sP-selectin.* *II*, *IV*, *VII*, *VIII*, *V Leiden*, *X*, *XIII*, *PF1.2 protein C*, *protein S*, *TAT*, *PIC*, *TM*, *t-PAIC.* *aPTT*, *PT*, *TT*, *VHAs.*	PE early diagnosis Risk stratification and PE progression. Monitoring anticoagulant and thrombolytic treatment. Identify DVT or cardiovascular diseases.
Cardiac biomarkers	*BNP*, *NT-proBNP*, *TnT*, *h-FABP*, *CK*, *myoglobin.*	PE early diagnosis Risk stratification and prognostic complications. Monitoring anticoagulant and thrombolytic treatment.
Inflammatory biomarkers	*NLR*, *PLR*, *LMR*, *CRP*, *TNF-α*, *IL-1β*, *IL-4*, *IL-6*, *IL-10 and plasma copeptin levels.* *Anticardiolipin*, *IgG and IgA.*	Identification of underlying inflammatory conditions. Monitoring systemic inflammation associated with PE. Prognosis and differentiation PE with higher inflammatory component vs PE with thrombotic features Monitoring thrombolytic treatment.
RNA biomarkers	*miR-210*, *miR-221*, *miR-222*, *miR-126-3p*, *miR-92a*, *and miR-132. miR-223*, *miR-145*, *miR-582*, *miR-195*, *miR-150*, *miR-21 and MiR-424*, *miR-134*, *miR-28-3p and miR-1233.*	Diagnostic and prognostic of PE. Identification of underlying PE pathophysiology. Future personalized treatment strategies.
	*NR_036693*, *NR_027783*, *NR_033766*, *NR_001284 and LncRNA-Ang362.* *hsa_circ_0002062*, *hsa_circ_0022342*, *hsa_circ_0016070*, *and hsa_circ_0046159.*	

Abbreviations: Soluble platelet selectin (sP-selectin), coagulation factors (*II*, *IV*, *VII*, *VIII*, *V Leiden*, *X*, *XIII*), prothrombin fragment 1 + 2 (PF1.2), antithrombin III complex (TAT), plasmin-α2-plasmininhibitor complex (PIC), thrombomodulin (TM), tissue plasminogen activator–inhibitor complex (t-PAIC), thromboplastin time (aPTT), prothrombin time (PT), thrombin time (TT), hemostatic assays (VHAs), b-type natriuretic peptide (BNP), troponin T (TnT), heart-type fatty acid binding protein (h-FABP), serum creatine kinase (CK), Immunoglobulin (Ig), neutrophil–lymphocyte ratio (NLR), platelet–lymphocyte ratio (PLR), lymphocyte–monocyte ratio (LMR), Serum C-reactive protein (CRP), alpha-Tumor necrosis factor (*TNF-α*), Interleukin (IL), microRNA(miR), Long non-coding RNAs (NR or lncRNA), and Circular RNAs (hsa_circ).

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
