# Peer review of "Decoding Pulmonary Embolism: Pathophysiology, Diagnosis, and Treatment"

_biomedicines, 2024, doi:10.3390/biomedicines12091936_

Round 1

Reviewer 1 Report (Previous Reviewer 2)

Comments and Suggestions for Authors

Authors significantly improved their manuscript. All parts of the manuscript are well written. 

Authors of this review summarized the contemporary knowledge about pathophysiology, diagnosis and treatment of pulmonary thromboembolism. Figure 1 is a very good diagnostic algorithm that should be used in everyday clinical practice. Section 5 gives a very comprehensive analysis of different biomarkers, although the cardiac biomarkers and d dimer are the only biomarkers used in clinical practice. . References cited in this manuscript are contemporary and relevant. I would recommend that part 7.6 be moved to the section on CTPE, considering that these are two related conditions.

I suggest that this systematic review can be accepted for publication.

Author Response

Reviewer 2 Report (Previous Reviewer 1)

Comments and Suggestions for Authors

I thank Peracaula et al for extensive revisions to their manuscript in response to previous comments. The 

manuscript is much better organized, easier to read, and clinically practical. As well, language issues and 

formatting and presentation have been improved. A few concerns persist, especially around some issues 

not addressed. 

CONCERNS: 

Table 1 – this has been improved, but confusion persists

- Left column for “clinical symptoms” should NOT include physical exam findings, eg. HR, O2sat; 

You could simplify by changing the title to “Clinical features”

- Unilateral leg swelling is not “predisposing factor”, but sign of potentially DVT, which of course 

can be associated with concurrent PE; I would move to left column 

Table 3 

- I don’t agree any of these biomarkers are “specific” for PE, as for example leukocytes, PLT, but 

also CK, TNF-alpha, etc. Please remove this word 

Figure 1 – this has not been corrected

- PE is still written as EP throughout diagram

Sections 5 / 6 – ED in PE vs Impact of PE in ED

- These 2 paragraphs remain poorly conceived. There is much overlap and confusion on whether 

ED can cause PE (uncommon but possible, largely in deep leg veins), and PE can cause 

pulmonary vascular ED. Needs revision. Would suggest keep the 1st section 5 for “General ED”, 

and then section 6 for ED in PE, and consodilate appropriate ideas in each section

Section 7 – very confusing use of terms/ explanations 

- VTS term is used but not defined / unclear; is this VTE? 

- 7.2: this section is not correct; the persistent symptoms described, regardless of the presence / 

absence of chronic PE on imaging, is post-PE syndrome. If indeed, imaging does confirm 

chronic PE, then the current term is chronic thromboembolic pulmonary disease (CTEPD); note 

this term is used in section 7.3, but not correctly defined. CTEPD is NOT simply new/recurrent 

PE, and it doesn’t make sense to say CTED predisposes to CTEPD

Author Response

Reviewer 3 Report (New Reviewer)

Comments and Suggestions for Authors

This well documented review focusses on an interesting topic, that of pulmonary embolism, including it's pathophysiology, clinical presentation, diagnostic approaches, and therapeutic interventions.

I have some suggestion for the authors:

1.      The affirmation "is treated into two main groups" should be rephrased.

2.      In Scoring systems, the authors mention that the Wells and Geneva scores confirm or exclude venous thromboembolism, which is incorrect. The presence of venous thromboembolism is determined by venous sonography. The 2 scores assess the risk for PE, not venous embolism and they do not establish the diagnosis for PE. The golden standard for this is CTPA.

3.      Regarding section Conventional anticoagulant Therapy, the author should add some clarifications. There is an acute phase, when the acute PE is detected, when intravenous non-fractionated heparin or subcutaneous low molecular weight heparin are recommended. Warfarin is used for the treatment of PE at discharge, not in the first hours after diagnosis.

4.      There is a third class of direct acting oral anticoagulant, inhibitors of anti-XI agents.

5.      In Supportive Care, the authors describe again the abbreviation of PE.

6.      I consider that the manuscript would benefit if the authors would structure it better. Pathophysiology should be at the beginning of the review, after the definition of PE.

7.      In my opinion the authors tend to mention at the beginning of each chapter basic, long known information, instead of making these sections shorter and promoting newer data. The result is a long, not very interesting review.

8.      I consider that it would help the review if the authors would add a chapter regarding treatment challenges. They could debate about the treatment options in patients with concomitant PE and haemorrhagic or ischemic stroke (Saleh Velez FG, Ortiz Garcia JG. Management dilemmas in acute ischemic stroke and concomitant acute pulmonary embolism: Case series and literature review. eNeurologicalSci. 2021 Apr 15;23:100341. doi: 10.1016/j.ensci.2021.100341. PMID: 33997324; PMCID: PMC8102755) and PE and acute aortic dissection (Tudoran M, Tudoran C. High-risk pulmonary embolism in a patient with acute dissecting aortic aneurysm. Niger J Clin Pract. 2016 Nov-Dec;19(6):831-833. doi: 10.4103/1119-3077.181355. PMID: 27811460).

9.      I recommend to mention the treatment after acute and chronic complications of PE.

10.  The conclusions are too long. I suggest to limit them to the important hypothesis only.

Comments on the Quality of English Language

Minor English editing

Round 2

Reviewer 3 Report (New Reviewer)

Comments and Suggestions for Authors They have answered each of my questions. I wanted to recommend the manuscript for publication Comments on the Quality of English Language

Minor English editing is needed.

Author Response

This manuscript is a resubmission of an earlier submission. The following is a list of the peer review reports and author responses from that submission.

Round 1

Reviewer 1 Report

Comments and Suggestions for Authors

Peracaula et al have submitted a narrative review of pulmonary embolism diagnosis and management. Specifically, they review ideas for potential prevention and diagnosis, optimal PE management, as well as some more recent research into relevant biomarkers that may improve PE diagnosis. The manuscript is comprehensive, but it can be hard to follow the ideas because of language issues and poor attention to formatting and presentation (eg. Table 1 title:  “This is a table”).  Overall, this is a useful article but could benefit from many clarifications, as well as some additions regarding the proposed areas to be covered.

MAJOR CONCERNS:

1.      Clinical value:  I am concerned that the paper is not as helpful clinically as it could be for many reasons. Besides poor attention to formatting and wording/grammatical errors, there are many statements which do not seem to be valid clinically, based on appropriate citations or clinical practice guideline. 

-        Moreover, many recommendations are vague, not providing any specific direction to clinicians (eg. Para 7.3:  “…these thrombolytic agents are utilized in different clinical contexts based on individual patient considerations”) ! This statement of course could apply to the entirety of medicine. 

-        The biomarker focus is interesting, but not as well developed as to be clinically useful.  Eg. Table 3:  This table may be more useful if a column is added to give a general description of the potential clinical uses of each category of biomarker.  Note cardiolipin and Ig’s would NOT typically be considered cardiac markers

-        There is confusion on the difference between the ‘clinical’ and ‘acquired’ risk factors, given there seems to be overlap between these two groups (ex. ‘Central venous lines’ as a clinical risk factor, and ‘indwelling venous catheters’ as an acquired risk factor; Paragraph 1 (PE).

2.      Specific issues about wording and idea clarity are presented below:   Eg. More concise language can be used in many areas- ex. 4.4, last paragraph of 6.2; First paragraph of 7.5, etc.

Paragraph 2 Diagnosis and Severity

-        Suggest including general overview of what symptoms and exam findings should raise suspicion for PE

-        Line 83: ‘it is essential to stratify patients based on their probability of developing a PE’- need to specify what patients (ie. Are these only patients with possible symptoms of PE, or all hospitalized patients?)

Table 1

-        Note incomplete title

-        Should read ‘Familial’ not familiar.

-        The  columns are poorly conceived or completed, with much  overlap between “Clinical history” and “clinical symptoms”; relatedly HR, sBP, and O2sat are not symptoms, but physical exam measures / signs

Figure 1

-        PE is written as EP throughout diagram- please correct

Paragraph 3: Endothelial Dysfunction in PE

-        Line 152: Suggest changing ‘elevated RV strain’ to ‘RV dysfunction’. The term ‘strain’ is used in echocardiography to describe how a chamber’s dimensions change over the cardiac cycle.

Figure 2:

-        Legend does not address the part of the diagram labelled ‘Altered vascular remodelling’

-        Note as well symbols (eg TFP) do not reconcile with legend abbrevs, and eNOS is not in the figure.

Paragraph 4.3

-        Beginning of this paragraph is unclear- it seems to say that endothelial dysfunction leads to endothelial dysfunction. Is it meant to start with ‘When the endothelium is damaged in PE...’?

-        What is the outcome of leukocyte adhesion and migration? 

-        Other similar statements are also circular, and require careful re-reading and editing

Paragraph 4.5

-        Line 220: ‘PH also influences left ventricular function...’- would change this to ‘severe RV dilation’ as this is typically what would cause issues with left heart function. Note as well the citation does not support this point, but instead is focused on PH in left-heart disease

-        Relatedly:  some of the basic references on PE are out of date, eg genetics of risks for PE where there has been extensive work since the 2009 Gohill paper.

Paragraph 5.1

-        Lines 254-256- if these tests have limited sensitivity and specificity, what makes them valuable tools for assessing PE?

Paragraph 5.2

-        Title should read cardiac biomarkers of PE (not EP)

-        Suggest specifying that you are discussing troponin T in line 262 when first mentioning Troponins.

-        Line 268- delete ‘diagnosing’

-        Line 274- suggest writing BNP/NT-proBNP as you would not run both of these tests together.

-        Note that improved prognostication with 2 factors, eg TnT and BNP would not be surprising, and could be helpful; this requires a citation

Paragraph 5.3

-        I am concerned that statements regarding the value of monitoring inflammatory markers (eg. ILs, CRP) in guiding PE “occurrence” and especially management are not clinically valid at present, as such testing is  not recommended by any clinical practice guidelines, not is standardly practiced

-         

Paragraph 5.4

-        Line 308, sentence ‘some lncRNAs have shown significant changes in pulmonary fibrosis....’- this sentence is a little unclear- do you mean the levels fluctuate in pulmonary fibrosis, or that the lncRNAs are associated with changes in the pulmonary fibrosis? Moreover, what is the relevance to your article on PE to mention specifically pulmonary fibrosis here?

Paragraph 6: Vascular diseases resulting from PH

-        Line 318-319: Would not use terms like ‘common’ or ‘often arise’ when describing CTEPH as this is considered a rare disease.

-        Line 321: Should read ‘cardiogenic’, not ‘cariogenic’

Paragraph 6.1: Recurrent thromboembolism disease

-        Title of paragraph should read ‘recurrent thromboembolic disease’

-        Line 329: change ‘coughing up blood’ to haemoptysis

Paragraph 6.2: Chronic thromboembolic pulmonary hypertension

-        Line 333-334: This sentence is unclear- should read as ‘characterized by persistently elevated PAP at least 3 months after initial PE diagnosis and effective anticoagulation’

Paragraph 6.3

-        Line 350: ‘many cases of PE’- A statistic here would be helpful

-        Post-thrombotic symptoms of PE is unclear; is this just typical post-DVT leg symptoms which can be seen in patients with PE in the absence of DVT (presumed initial DVT), or do the authors mean to discuss post-PE syndrome?

Paragraph 6.4: RV Failure

-        Line 367: Change ‘edema, often observed in as swelling in the legs and ankles,’ to ‘peripheral edema’

-        Line 370: Need to specify that anticoagulation is a potential treatment for RV failure only secondary to PE, as it is not a treatment for RV failure from other causes.

Paragraph 6.5: Cardiogenic Shock

-        The second paragraph in this section repeats much of what was discussed in the previous ssection about RV failure. There should be a specific focus on management of cardiogenic shock here (ex. Inotropes).

Paragraph 7.1: Conventional Anticoagulation Therapy

-        Line 397: ‘endothelial dysfunction’ should be ‘ED’

Paragraph 7.5: Supportive Care

-        Line 481: What do you mean by ‘individualized oxygen therapy regimens’?

Comments on the Quality of English Language

As in attached file.  Significant revision required

Reviewer 2 Report

Comments and Suggestions for Authors

This is a very good systematic review. Authors of this review summarized the contemporary knowledge about pathophysiology, diagnosis and treatment of pulmonary thromboembolism.

All parts of the manuscript are well written. In the first section authors gave very good explanation of „massive“ and „non-massive“ PE, because sometimes clinicians and radiologist have different definition of „massive“ PE. Figure 1 is a very good diagnostic algorithm that should be used in everyday clinical practice. Section 5 gives very comprehensive analysis of different biomarkers, although the cardiac biomarkers and d dimer are the only biomarkers used in clinical practice. Understanding the mechanisms of vascular diseases resulting from PE can help in making therapeutic decisions (for example, thrombolytic therapy for prevention of CTEPH in patients with intermediate-high risk PE and low-bleeding risk). References cited in this manuscript are contemporary and relevant.

I suggest that this systematic review can be accepted for publication.

 Thank you

Round 2

Reviewer 1 Report

Comments and Suggestions for Authors

We appreciate the reviewers attempt to respond to our and other reviewer’s previous suggestions so quickly Significant editing is still needed for clarity, grammar and especially accurate and clinically-relevant content  

MAJOR CONCERNS: 

I remain concerned about the usefulness of this review.  As per the idea behind the paper ... “…synthesizing existing knowledge, the review provides a comprehensive outlook of enhance PE management and preventive strategies to clinicians and researchers”.  The current manuscript as a narrative review does not do this, eg. Not comprehensive but picks out some ideas and leaves out other important ones in the PE field; no clear summary of management of acute PE or of issues post-PE, eg. Post-PE syndrome or CTEPH, no clear “preventive strategies” that I can identify. Moreover, the information provided is often inaccurate, eg. In situ thrombosis is NOT typically a feature of acute PE, CTEPH is NOT a common complication of PE, the proposed biomarkers are NOT currently used to follow patients with acute PE, etc; as detailed below.  Moreover, the discussions of pathophysiology are incomplete and hard to follow.  

In the current state, the article does not flow well. It begins with a heavy clinical focus, then transitions into some pathophysiology/cell biology, then transitions back into clinical care. It is clear that the topic proposed to be covered is quite broad and the authors could consider focusing on one aspect of PE more completely / rigorously - either diagnosis/management or the pathophysiology/biomarkers. My suggestion would be to remove much of the content on diagnosis/management of PE (as little new information is added to this topic), and better develop the content on pathophysiology and biomarkers.  

Specifically regarding the biomarkers section, simply listing biomarkers linked to PE is not helpful. For example, line 252: In addition, coagulation factors such as II, IV, VII, VIII, V Leiden, X, XIII, prothrombin fragment 1+2, antithrombin III, tissue factor, protein C, and protein S have also been linked to PE’; is this as simple as the fact that thrombosis requires activation of all the clotting cascade factors, and the natural inhibitors prevent thrombosis, so that their deficiency, long-recognized, contributes to thrombosis? Line 290: “In addition, anticardiolipin antibodies IgG and IgA are linked to PE”. Statements like this are not clinically useful, nor do they provide enough information to even encourage further investigation in a research setting.  

Other concerns are discussed below by section:  

Section 1: Pulmonary Embolism 

  • The definitions of PE in this paragraph seem to be from the European Society of Cardiology’s 2019 guidelines- if so, these guidelines should be cited.  

  • Line 57: States that hemodynamically unstable PEs were previously referred to as ‘high-risk’ PEs. The ESC guidelines seem to use ‘high-risk PE as current terminology. Is there another guideline you are using? If so, this should be cited. 

  • Line 63: Similar to above, low-risk PE is still an actively used term throughout these guidelines. Further, per the table below (taken from the ESC guidelines), a hemodynamically stable PE is not equivalent to a low-risk PE.  

Section 2:  

  • Line 79: ‘primarily non-invasive' - all the diagnostic techniques you discuss are non-invasive, so this is not necessary to include.  

  • Line 120: Table 2 uses ‘Moderate’ rather than ‘intermediate’. The term used should be kept consistent for clarity.  

Table 1 

  • There still seem to be inconsistencies in what are listed as ‘clinical factors’ here vs what are listed as ‘clinical factors’ in the text (line 47) Given the subsequent risk score systems are completely presented, ideally would remove this table which provides no valueAlternatively, could be kept but needs consistency for more clarity.  

Section 3: Endothelial Dysfunction 

  • Line 146: Remove word ‘external’- genetic susceptibility is not an ‘external’ risk factor.  

Section 4: Consequences of ED in PE 

  • Line 170: This sentence does not make sense. Please rewrite.  

Section 6: Vascular diseases resulting from PE 

  • Line 327: remove ‘frequently’. Ie. ‘diseases that can be observed after PE...’ illnesses like CTEPH and cardiogenic shock are not ‘frequent’ complications of PE.  

  • Section 6.3: If including post-thrombotic syndrome as a consequence of PE, what is the reason for not including diseases such as chronic thromboembolic pulmonary disease (CTED or CTEPD) or post-PE syndrome? 

  • Section 6.4: New sentence added in line 381 is not necessary. Rather, line 380 could read ‘Typical treatments for RV failure secondary to PE....” 

  • Section 6.5: Line 397- Coronary revascularization and CABG for ACS is related to left heart disease and not related to PE. This should be removed as it is irrelevant. The acute management of RV failure in setting of PE is very different that stated   

  • Line 401: Digoxin is not used in acute cardiogenic shock, unless rarely complicated by atrial arrhythmias  

MINOR CONCERNS 

There remain numerous grammatical errors, for example: 

  • Line 67: patients who present with right ventricular (RV) dysfunction 

  • Line 114: The diagnosis of PE can involve D-Dimer tests...” 

  • Line 206: “chain reaction involving CAMs that will...’ 

  • Line 402: “and are often used in” 

  • Line 471: “These thrombolytic agents address severe PE by...” 

This is not an exhaustive list as there are numerous other ones that will take careful English-language review and editing

Comments on the Quality of English Language

Extensive editing is required to improve clarity and readability